# Interactive Effects of Scion and Rootstock Genotypes on the Root Microbiome of Grapevines (*Vitis* spp. L.)

Stefanie Nicoline Vink [1,*], Francisco Dini-Andreote [1,2], Rebecca Höfle [3], Anna Kicherer [3] and Joana Falcão Salles [1,*]

1   Groningen Institute for Evolutionary Life Sciences, University of Groningen, 9747AG Groningen, The Netherlands; fjd5141@psu.edu
2   Department of Plant Science & Huck Institutes of the Life Sciences, The Pennsylvania State University, University Park, State College, PA 16802, USA
3   Institute for Grapevine Breeding Geilweilerhof, Julius Kühn-Institut, 76833 Siebeldingen, Germany; rebecca.hoefle@julius-kuehn.de (R.H.); anna.kicherer@julius-kuehn.de (A.K.)
*   Correspondence: vinksn@gmail.com (S.N.V.); j.falcao.salles@rug.nl (J.F.S.); Tel.: +31-50-363-2162 (J.F.S.)

**Abstract:** Diversity and community structure of soil microorganisms are increasingly recognized as important contributors to sustainable agriculture and plant health. In viticulture, grapevine scion cultivars are grafted onto rootstocks to reduce the incidence of the grapevine pest phylloxera. However, it is unknown to what extent this practice influences root-associated microbial communities. A field survey of bacteria in soil surrounding the roots (rhizosphere) of 4 cultivars × 4 rootstock combinations was conducted to determine whether rootstock and cultivar genotypes are important drivers of rhizosphere community diversity and composition. Differences in α-diversity was highly dependent on rootstock–cultivar combinations, while bacterial community structure primarily clustered according to cultivar differences, followed by differences in rootstocks. Twenty-four bacterial indicator genera were significantly more abundant in one or more cultivars, while only thirteen were found to be specifically associated with one or more rootstock genotypes, but there was little overlap between cultivar and rootstock indicator genera. Bacterial diversity in grafted grapevines was affected by both cultivar and rootstock identity, but this effect was dependent on which diversity measure was being examined (i.e., α- or β-diversity) and specific rootstock–cultivar combinations. These findings could have functional implications, for instance, if specific combinations varied in their ability to attract beneficial microbial taxa which can control pathogens and/or assist plant performance.

**Keywords:** agricultural practices; cultivar; grafting; interaction rootstock scion; plant performance; rhizosphere bacteria; taxonomic indicators; viticulture





## 1. Introduction

Soil microbial communities perform a wide range of ecosystem processes and functions and are increasingly recognized as vital components of a healthy agroecosystem. For instance, bacteria play pivotal roles in the biogeochemical cycling of nutrients, and influence plant productivity and health through the action of specific plant growth-promoting (PGPB) and biocontrol bacterial species, and/or negatively, through the actions of plant pathogens [1,2]. Rhizosphere bacterial diversity is known to affect plant health, with communities with a higher diversity generally better able to withstand invasion of pathogens and possessing higher amounts of PGPBs [3–5]. The higher biodiversity often relates to increasing levels of ecosystem functions and services [6,7]. In addition, plant functional traits, such as abiotic and biotic tolerance (e.g., salinity, drought, diseases) and nutrient uptake, are to varying degrees directly influenced by the root-associated microbial communities [8–10].

Conversely, soil communities are also affected by plant characteristics, primarily through the production of root exudates. For instance, up to 40% of the photosynthates produced by a plant can be actively released by the roots [11], and the quality and quantity

of these carbon compounds can vary between and within plant species [12–14], potentially leading to highly specific relationships between plants and microorganisms [15,16], such as those between rhizobia and legumes [17] or with plant pathogens [18,19]. Bacterial communities are also known to differ according to plant genotypes and hosts [12,15,20,21], although this effect is by no means ubiquitous, and plant genotypic differences do not always lead to significant differences in microbiomes in the rhizosphere [22].

Viticulture is one of the world's main horticultural practices. In 2018, approximately 77.8 million tons of grapes were produced globally, primarily for wine production [23]. Grafting is common practice in viticulture to reduce the incidence of the grapevine pest phylloxera; European *Vitis vinifera* L. scions (i.e., the upper plant) merged with North American *Vitis* sp. hybrid rootstocks [24]. This results in plants with stress- and disease-resistance combined with desirable agronomic characteristics for grape production and allows breeders to select specific traits independently for rootstocks and cultivars.

There is a considerable body of research on the effect of rootstock on scion physiology and physical properties in viticulture, for instance, by showing improved drought tolerance [25,26], and changes in stomatal conductance and transpiration [27]. Less research has been conducted on the effects of the scion on the rootstock, and, to date, this effect has been examined only on physical rootstock properties such as root biomass and length [28]. The interaction between rootstocks and scions also remains understudied, but research has shown that it can affect yield and quality of grapes [29], and root behavior and plant growth [30]. With grafting, scions and rootstocks each maintain their own genetic identity [24], and in other plant species this has led to changes in bacterial diversity [31] or community structure [32].

Despite the potential influence of grapevine genotypic variation on bacterial community structure and functioning in the rhizosphere, and the importance of the root microbiome for plant development and health, little is known about how rootstock and scion, and importantly, their interaction, exert an influence on soil bacterial communities in grapevines. Therefore, a survey was conducted with the aim of elucidating the effect of different rootstock and grafted scion combinations on soil bacterial communities associated with the rhizosphere of grapevines.

## 2. Materials and Methods

### 2.1. Experimental Design and Soil Sampling

In October 2016, a field survey of bacteria from the rhizosphere of different scion-rootstock combinations was conducted. Grapevines were planted 11 years prior to the experiment in a vineyard at the Institute for Grapevine Breeding, Julius Kühn-Institut in Siebeldingen, Germany. The vineyard was located in an area with sandy loam soil (type pararedzina). In 2016, the area received approximately 625 mm rainfall.

Each of four scion cultivars (Calandro, Reberger, Felicia, and Villaris; from now on referred to as cultivars) were grafted onto each of four rootstock types (125 AA, 5 BB, Binova, and SO 4). All cultivars belong to the *Vitis vinifera* species; Calandro and Reberger are red grape varieties, and Felicia and Villaris are white grapes. The rootstocks were all *Vitis berlandieri* and *Vitis riparia* crosses with low genetic variation [33,34]. Grafted grapevines were placed in adjacent rows which were all subjected to cover cropping: every second row with a vineyard-specific mixture of various legumes and other forbs (Wolff-mix) and tilled via disc harrow in spring each year, and the remaining rows with grass and mulched three times between April and August, depending on the weather and growing conditions. Fertilization was conducted yearly with Entec 26 (26% N, 13% S) at a rate of 39 kg N/ha. For the duration of the experiment no irrigation took place.

Rhizosphere soils were collected in a single day, from three randomly selected plants from each cultivar x rootstock combination, yielding a total of 48 samples. Samples were collected by removing roots with attached soil in the field using sterile metal forceps, cleaned with 70% ethanol between each sample to avoid cross-contamination. Samples were transported on ice and processed in the laboratory within 24 h. Sampled roots were

shaken to remove loosely adhering soil, and rhizosphere soil was collected by removing the remaining firmly attached soil particles using a sterile disposable brush. Samples were stored at $-20°$ C for further DNA extraction.

### 2.2. DNA Extraction and Sequencing

DNA was extracted using the PowerSoil DNA extraction kit (Qiagen, Hilden, Germany) following the manufacturer's instructions, and 0.5 g of rhizosphere soil per sample, with an addition of 50 mg of sterile glass beads (ø 0.1 mm, BioSpec Products, Bartlesville, WA, USA) to improve extraction yield. DNA quality and quantity were determined using a NanoDrop 2000 Spectrophotometer (Thermo Fisher scientific, Walthan, MA, USA). Amplicon library preparation and sequencing based on the V4 region of the bacterial 16S rRNA gene was conducted at the Argonne National Laboratory Sequencing Facility (Lemont, IL, USA) using the Illumina MiSeq platform, following the Earth Microbiome Project protocol [35], yielding paired-end reads of 150 bp in length.

### 2.3. Sequence Quality Filtering and Data Analysis

Reads were imported into QIIME2 version 2018.2 [36], demultiplexed and primers were removed using the EMP procedure. Sequences were filtered, dereplicated and denoised, chimeric sequences were removed, and paired using the DADA2 plugin [37] in QIIME2, resulting in amplicon sequence variants (ASVs) of 253 bp in length. ASVs were aligned using MAFFT [38] and used to generate a mid-point rooted phylogenetic tree using FastTree [39]. Taxonomic assignment was performed using the RDP classifier version 2.10.1 [40]. Full details and settings are presented in the Supplementary Materials.

All statistical analyses were conducted in R version 3.6.2 [41]. The ASV table, taxonomy table, phylogenetic tree and associated metadata were imported into the package Phyloseq version 1.30.0 [42], and any non-bacterial ASVs and those taxonomically affiliated to chloroplast and mitochondrion were removed. The resulting table was rarefied to 15,704 sequences per sample to account for any differences in sequencing depth.

Observed ASV richness, Shannon and Simpson diversity indices were calculated using Phyloseq and Faith's phylogentic diversity (PD) in the packages picante version 1.8 [43] and btools version 0.0.1 [44]. Significant differences in $\alpha$-diversity measures (i.e., variation of species within a sample) between cultivars, rootstocks and their interaction were tested using ANOVA on Aligned Rank Transformed data (i.e., a non-parametric ANOVA) using the package ARTool version 0.10.7 [45]. This method was used because the $\alpha$-diversity measures did not meet the assumption of normally distributed and/or homogeneity of variance for a parametric ANOVA. As PD showed the strongest effect of treatment, a post hoc analysis was performed for the rootstock $\times$ cultivar interaction term using estimated marginal means with the package emmeans version 1.4.5 [46].

Bacterial community structure ($\beta$-diversity, i.e., variation in species between samples) was assessed using principal coordinate analysis (PCoA) based on Bray-Curtis distances, which calculates dissimilarity between samples using relative sequence abundances. To determine potential significant effects of rootstock, cultivar and their interaction, PERMANOVA with 999 permutations was used with the package vegan version 2.5.4 [47]. Bacterial taxonomic composition was evaluated by merging taxa to the phylum level (and summing their abundances), and removing those with an abundance <20. Data were plotted based on relative abundances per rootstock and cultivar combination. To detect bacterial taxa with an affinity towards specific (combinations of) cultivars or rootstocks, indicator species analysis using the package indicspecies [48] was performed using abundance data of taxa merged at the genus level. The analysis was carried out separately for rootstock and cultivar samples and resulted in a list of species associated with individual, or groups of, cultivars or rootstocks, based on calculations that take into account the strength of association and the statistical significance of the relationship between species abundances and treatment groups.

## 3. Results

### 3.1. Bacterial α-Diversity across Different Cultivar-Rootstock Combinations

Rootstock identity was found to exert a significant influence on rhizosphere bacteria for all α-diversity measures, while cultivar identity only significantly affected Simpson and Faith's PD indices (Table 1). However, all α-diversity measures showed a significant interaction between cultivars and rootstocks (Table 1) indicating that the observed differences between rootstocks varied in strength according to cultivar identity.

**Table 1.** ANOVA results of α-diversity measures per rootstock and cultivar identity and the interaction between the two factors. Data were aligned rank transformed prior to statistical analysis. Significance levels are as follows: *** $p < 0.001$, ** $p < 0.01$, * $p < 0.05$, ns = not significant.

|  | Richness | Shannon | Simpson | Faith's PD |
|---|---|---|---|---|
| Rootstock | $F_{3,32} = 4.6$ ** | $F_{3,32} = 5.2$ ** | $F_{3,32} = 3.2$ * | $F_{3,32} = 7.3$ *** |
| Cultivar | $F_{3,32} = 0.9$ ns | $F_{3,32} = 2.6$ ns | $F_{3,32} = 4.5$ ** | $F_{3,32} = 7.1$ *** |
| Rootstock:Cultivars | $F_{9,32} = 3.9$ ** | $F_{9,32} = 4.0$ ** | $F_{9,32} = 3.0$ ** | $F_{9,32} = 4.8$ *** |

Of all α-diversity measures, Faith's PD showed the largest difference for both rootstocks (Figure 1a) and cultivars (Figure 1b) as main effects, in addition to their interaction (Figure 1c). This interaction effect was also observed for the other diversity measures (Figure S1). Post hoc analysis on Faith's PD indicated that the most distinct rootstock–cultivar combinations were 5 BB-Reberger, 125 AA-Calandro and 125 AA-Vilaris. However, values of Faith's PD were quite similar for the majority of cultivar x rootstock combinations.

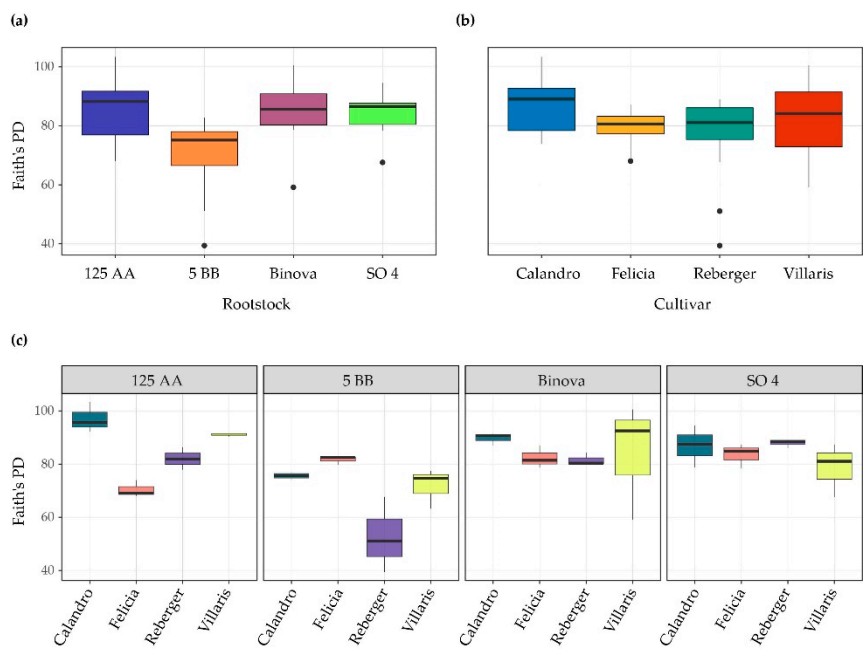

**Figure 1.** Faith's phylogenetic diversity of bacterial rhizosphere communities per (**a**) cultivar, (**b**) rootstock and (**c**) the interaction between rootstock and cultivar.

### 3.2. Bacterial β-Diversity across Different Cultivar-Rootstock Combinations

The main determinant of bacterial community structure in the rhizosphere was cultivar identity (PERMANOVA: $R^2 = 0.14$, $p < 0.001$), with samples from Calandro and Villaris plants harboring a different community to those of Felicia and Reberger (Figure 2a).

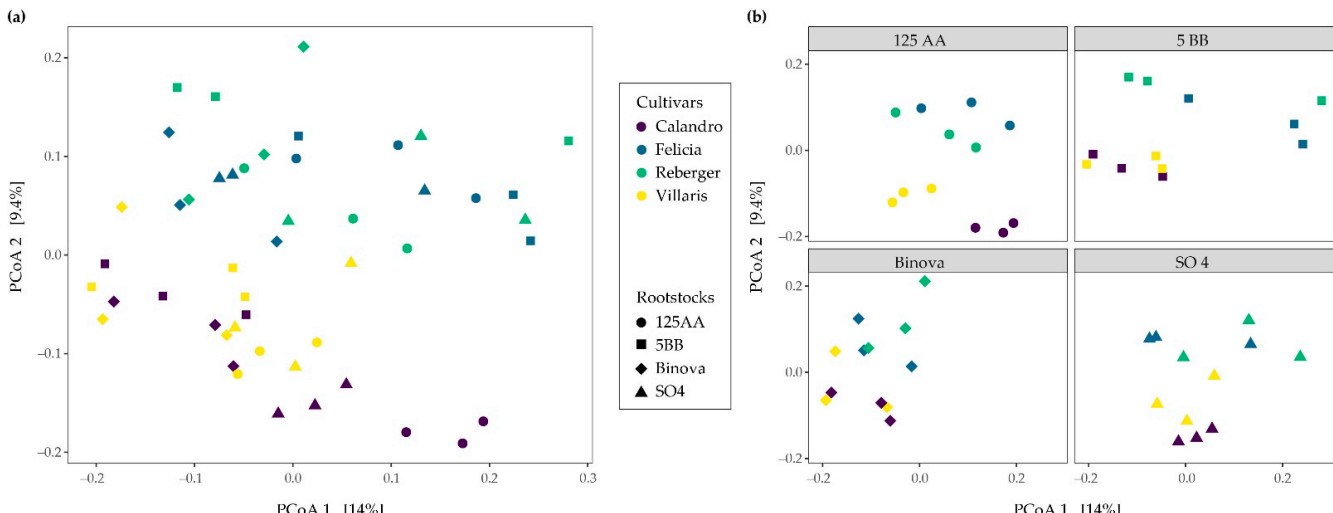

**Figure 2.** PCoA of rhizosphere soil bacterial communities based on the Bray-Curtis dissimilarities. The figure displays (**a**) collection of all samples or (**b**) plots separated according to each rootstock.

Although this separation was independent of rootstock identity, i.e., there was no significant interaction effect (PERMANOVA: $R^2 = 0.18$, $p = 0.65$), in particular the combination 125 AA-Calandro and 5BB-Calandro harbored a distinct bacterial community from the other cultivar–rootstock combinations (Figure 2a). Rootstock identity was found to exert a comparatively minor significant effect on differences in bacterial community structure (PERMANOVA: $R^2 = 0.10$, $p < 0.001$). However, such differences were less visible than that imposed by differences in cultivar identity within each rootstock.

### 3.3. Taxonomic Level Analysis

Taxonomic composition at the phylum level did not vary greatly between the different cultivar–rootstock combinations (Figure 3). For all combinations, the phyla of *Proteobacteria*, *Acidobacteria*, *Actinobacteria*, *Bacteroidetes* and *Firmicutes* had the highest relative abundance, and while the abundances of the less dominant phyla varied per rootstock–cultivar combination, here there were also no large differences observed.

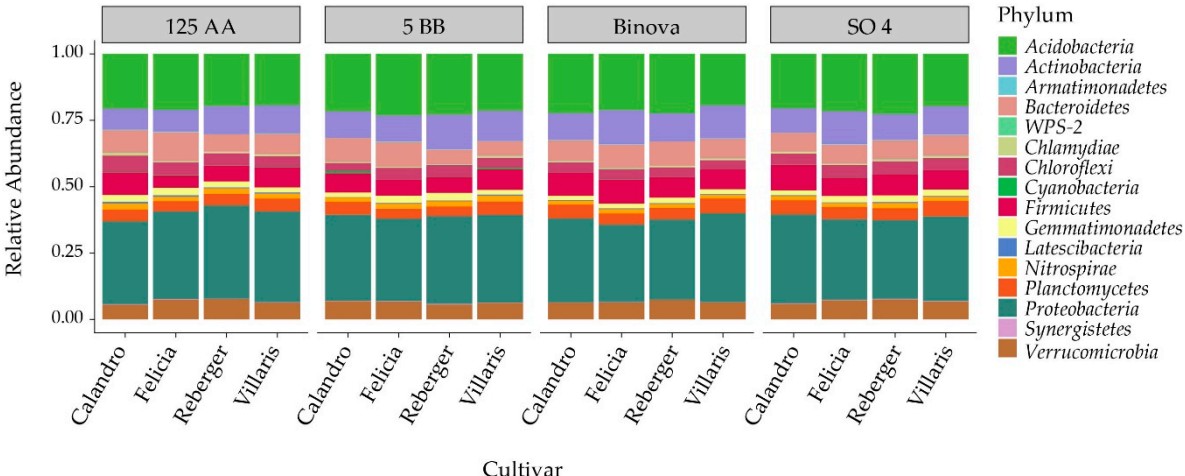

**Figure 3.** Taxonomic composition of rhizosphere bacterial communities per rootstock and cultivar. Bars show the relative abundance of bacterial taxa merged at the phylum level. Phyla whose abundance was <20 were not shown for clarity of data presentation.

### 3.4. Bacterial Indicator Species Analysis

Indicator species analysis was conducted separately for cultivars and rootstocks at the genus level. Twenty-four bacterial genera were associated with one or more cultivars (Figure 4a), while only thirteen were found to be specifically associated with the rootstocks (Figure 4b), and only one genus (*Elioraea*) was associated with both rootstock and cultivars. The number of genera that associated with a single cultivar or rootstock were low-*Marmoricola* was specifically associated with one cultivar (Reberger) while the genera *Escherichia/Shigella* and *Enterobacter* were exclusively associated with the rootstock 125 AA. Instead, the vast majority of bacterial genera were indicative of multiple groups, with the majority associated to 3 (out of the 4) groups for both cultivars (Figure 4a) and rootstocks (Figure 4b).

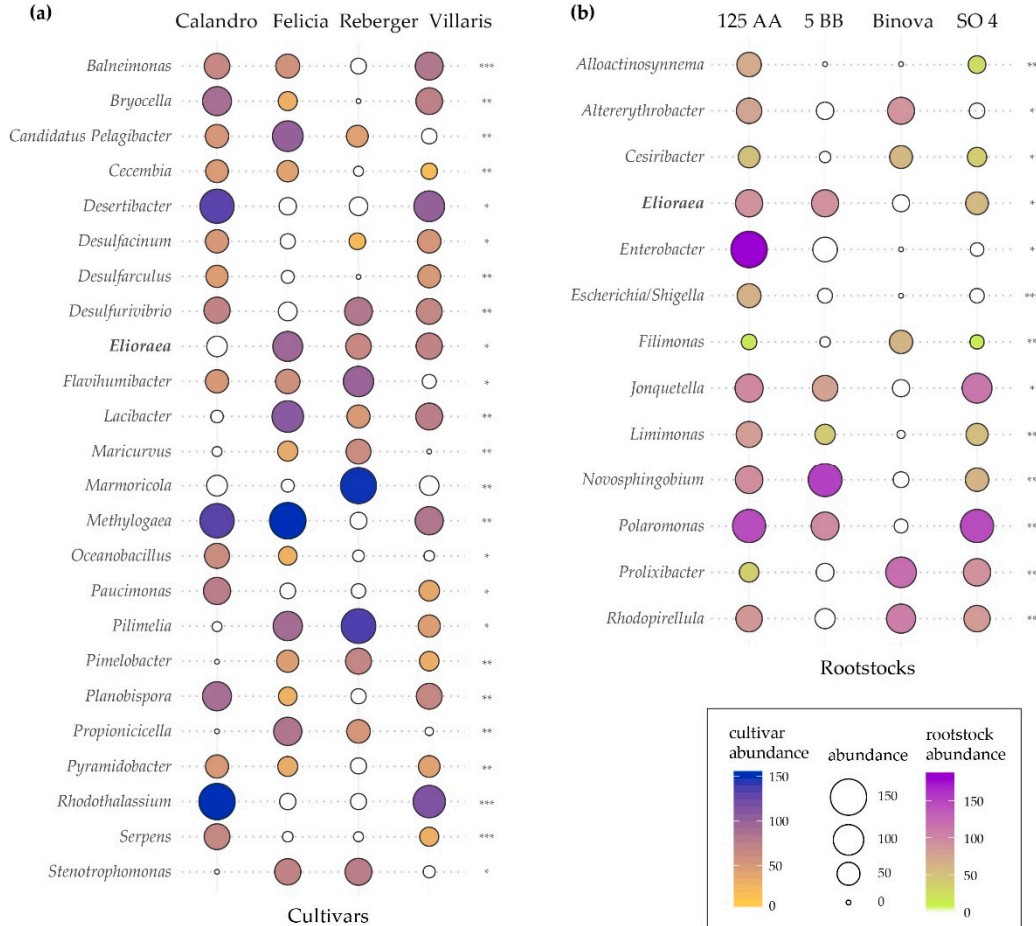

**Figure 4.** Indicator species analysis of bacterial genera per cultivars (**a**) or rootstocks (**b**). Color and bubble size relate to sequence abundance per genus, bubbles without color indicate that the genus was not an indicator for that particular cultivar or rootstock. *** $p < 0.001$, ** $p < 0.01$, * $p < 0.05$. Genera that are indicator species in both datasets are shown in bold.

The genus *Elioraea*, which was an indicator species in both rootstocks and cultivars, is relatively unknown, with few known members; the genus is thought to belong to the aerobic anoxygenic photosynthetic bacteria [49], which are obligate aerobes that can capture energy from light through photosynthesis [50]. Species in the genus *Marmoricola* are all mesophilic, non-pathogenic and often found in soil environments [51]. The genera *Escherichia/Shigella* and *Enterobacter* all belong to the Enterobacteriaceae, and have members which are known human, animal and plant pathogens, as well as plant-associated growth-promoting species, and lignin degraders [52]. The genera *Desulfacinum*, *Desulfarculus* and *Desulurivibrio*, which were indicator species for all but the cultivar Felicia, are known to be anaerobic sulfate reducers [53].

## 4. Discussion

The aim of this study was to determine whether rootstock and cultivar identity are important drivers of the structure and composition of rhizosphere communities in grapevines. Based on the results from experiments on other plant species [31,32] and the limited knowledge in grapevines [9,54], the expectation was that there would be an effect of grafting on rhizosphere microbial communities of different cultivar–rootstock combinations. For many of the measures that were examined, such as α-diversity and taxonomic composition, highly specific cultivar–rootstock interaction effects were observed, but on the whole, these effects occurred for a few specific rootstock–cultivar combinations.

However, a clear differentiation in bacterial community structure (β-diversity) was observed, and in particular a clear distinction was observed between two groups of cultivars; Calandro and Villaris on the one hand and Felicia and Reberger on the other. There is clear genetic component to plant root exudate composition [22,55,56] and this can strongly affect rhizosphere bacterial communities in *Vitis* sp. [57]. The two red cultivars (Calandro and Reberger) were more closely related to each other (sharing one parent) than to the two white grapevine cultivars, which were also closely related (sharing both parents) [34]. Therefore, the expectation was that there would be a differentiation along red and white cultivars, but this was not the case. Due to the genetic basis of root exudate composition, examination of root exudate quality, and its effect on bacterial communities, might offer more insight into the causes of the observed effect.

Less difference was observed between the different rootstock genotypes. While the cultivars consisted of genotypes from very different parents, the rootstocks all had a similar lineage [33] and therefore were likely to be more similar genetically (and phenotypically) than the cultivars. Previous studies have shown clear differences in bacterial communities between different rootstock genotypes grafted with a single scion genotype [54]. However, this can be context dependent, and rhizosphere bacterial communities have been shown to differ between rootstocks in one vineyard and not in another [58]. In addition, Berlanas et al. [58] found no difference in rhizosphere microbiomes between different 7-year-old grapevine rootstock genotypes, but that mature (25 years old) rootstock genotypes were associated with different rhizosphere microbiomes. In this context, given that the grapevines used in the current study were 11 years old, the specific effect of rootstock genotype might not have yet developed. Many aspects of both host and environment are responsible for the quality and quantity of root exudates [55], and hence in shaping rhizosphere bacterial communities, and it seems likely that a combination of factors played a role in shaping bacterial communities in the current experiment.

The α-diversity measures were determined to a large degree by the interaction between rootstock and cultivar. Although most combinations did not differ significantly, there were some notable exceptions, such as the combinations 5 BB-Reberger and 125 AA-Felicia with low, and 125 AA-Calandro with high α-diversity values. This variation could have functional implications as microbial species richness has been linked to resistance to invasion of pathogens [3,4]. High species richness, particularly combined with a more even distribution of those species, is often paired with improved ecosystem services and stability, and enhanced plant performance due to beneficial traits [59–61]. In the current study, the strongest response was observed with Faiths Phylogenetic Diversity, implying a potentially higher diversity in terms of functions [62] and resistance to invasion [63].

There were no large differences between any of the rootstock–cultivar combinations at the phylum level in the current study. The rhizosphere soils were dominated by the phyla Proteobacteria, Acidobacteria, Actinobacteria, Bacteriodetes and Firmicutes, which is in line with soils found across the globe [6], as well as in vineyards [64,65]. However, functional effects of the microbiome on plants are more likely to occur at lower taxonomic levels, and individual taxa can also make a large impact, such as with biocontrol, PGP and pathogenic bacteria [1,2]. Indicator species analysis showed that, on the whole, genera were differentially abundant in more than one cultivar or rootstock genotype, and at least some of these genera are known to differ in terms of their functionality. Five bacterial

genera were in relatively higher abundance in the rhizosphere of the cultivars Calandro and Villaris, which grouped together in terms of bacterial community structure, than in the other two cultivars. For this cultivar group, two genera *Desertibacter* and *Rhodothalassium* occurred in relatively high abundance. Both of these genera belong to families [66,67] with members known to be able to benefit plant growth [68]. Three genera were specifically associated with the other β-diversity cultivar group (Felicia and Reberger), but here, only one genus (*Stenotrophomonas*) was in high abundance. Species in the *Stenotrophomonas* are known to be closely associated with plants, and also have known plant beneficial effects [69]. Although speculative, it seems that both groups of cultivars are able to recruit (i.e., attract and select) different, potentially beneficial, genera.

## 5. Conclusions

The results from this study reveal for this first time that bacterial diversity in grafted grapevines is affected by both cultivar and rootstock identity, but that this effect is dependent on which diversity measure is being examined (i.e., α- or β-diversity) and specific rootstock–cultivar combinations. This finding is of particular importance since it could affect the ability of specific combinations to chemically attract, via root exudates, specific bacteria that could affect key functions associated with plant protection, performance, and productivity [70], as well as specifically to viticulture because it can affect colonization in other plant organs such as grapes [65], potentially impacting wine quality [71]. A first step toward further understanding the nature and cause of the observed rootstock x scion interactive effects on soil bacterial communities in this study would be to unravel the link between grapevine genomes with their root exudation patterns, as well as to their rhizosphere bacterial communities.

**Supplementary Materials:** The following are available online at https://www.mdpi.com/2076-3417/11/4/1615/s1, Figure S1: observed richness, and Shannon and Simpson diversity indices of bacterial rhizosphere communities per rootstock and cultivar interaction.

**Author Contributions:** Conceptualization, F.D.-A. and J.F.S.; Data curation, S.N.V.; Formal analysis, S.N.V.; Funding acquisition, J.F.S.; Investigation, F.D.-A., R.H. and A.K.; Methodology, F.D.-A.; Project administration, S.N.V., F.D.-A. and J.F.S.; Resources, R.H. and A.K.; Software, S.N.V.; Supervision, J.F.S.; Visualization, S.N.V.; Writing—original draft, S.N.V.; Writing—review and editing, S.N.V., F.D.-A., R.H., A.K. and J.F.S. All authors have read and agreed to the published version of the manuscript.

**Funding:** This research was co-financed by the European Union (FACCE SURPLUS ERA-NET grant 652615) and the Netherlands Organization for Scientific Research (NOW—grant number ALW.FACCE.1), as part of the international collaborative 'Vitismart' project.

**Institutional Review Board Statement:** Not applicable.

**Informed Consent Statement:** Not applicable.

**Data Availability Statement:** The sequencing reads from this study are openly available in the European Nucleotide Archive under accession nr. PRJEB42324. This data can be found at https://www.ebi.ac.uk/ena/browser/view/PRJEB42324 (accessed on 20 January 2021).

**Acknowledgments:** The authors would like to thank Jolanda Brons for assistance with sampling and Jan Veldsink for assistance with molecular methods. We would also like to thank the Center for Information Technology of the University of Groningen for their support and for providing access to the Peregrine high-performance computing cluster.

**Conflicts of Interest:** The authors declare no conflict of interest. The funders had no role in the design of the study; in the collection, analyses, or interpretation of data; in the writing of the manuscript, or in the decision to publish the results.

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
