# Peer review of "Interactive Effects of Scion and Rootstock Genotypes on the Root Microbiome of Grapevines (Vitis spp. L.)"

_applsci, doi:10.3390/app11041615_

Round 1
Reviewer 1 Report
A partly annotated version of the manuscript is uploaded to specify modifications and clarifications needed. Morevoer main issue to be explained is the relevance of the results for the field that is not clear moreover the personalization form must be avoided in the whole manuscript.

Reviewer 2 Report
This is an incredibly interesting and important study. Additional comment on long-term management conditions of the vineyard site would be helpful to gain a bit more context. Additionally, environmental data (climate) would be useful to as abundance and diversity has been shown to be tied to this.
Reviewer 3 Report
The researchers examined the root microbiome of grafted grapevines derived from combinations of four scions (cultivars) and four rootstocks. The Introduction section is quite informative and sufficiently describes the scope of the research, while the objective is clearly stated. Materials and methods require some important information regarding the experimental design and some other factors.
Specific comments are following.
- Line 27. Keywords should not be included in the title. Please remove or substitute “rootstock” and “scion”
- Line 39. I suggest that “(a)biotic” is changed to “biotic and abiotic”
- Line 53. Do you mean 7.4 million throughout the world? Please specify in the text.
- Lines 78-80. This part belongs to the Discussion section.
- Line 82. Please add more information about the experimental design and other important factors. For example, soil type, fertilization, watering patterns and other relevant information.
- Line 93. Please check if this is correct. If I understand correctly you tested 16 cultivar-rootstock combinations and 3 plants per combination, thus 48 samples. Did you also use non-grafted scion and rootstock cultivars?
- Line 116. I suggest that the supplementary figure is included as a regular figure in the manuscript since it shows important information.
- Lines 236-237. Do you have a theory about these results?
- Line 265. Please add a comment about taxonomic level analysis even though no significant differences were observed between the cultivar-rootstock combinations. Maybe there is an explanation in the literature.
- Line 275. You may include possible future research objectives.
Round 2
Reviewer 1 Report
A partly annotated version is enclosed with comments and suggestions.
